# Investigating Aluminum Tri-Hydroxide Production from Sodium Aluminate Solutions in the Pedersen Process

James Malumbo Mwase [1,*], Michail Vafeias [2], Danai Marinos [2], Panias Dimitrios [2] and Jafar Safarian [1]

1 Department of Materials Science and Engineering, Norwegian University of Science and Technology (NTNU), 7491 Trondheim, Norway; jafar.safarian@ntnu.no
2 School of Mining and Metallurgical Engineering, National Technical University of Athens, 15780 Athens, Greece; michalisvafeias@gmail.com (M.V.); danaimarinou27@gmail.com (D.M.); panias@metal.ntua.gr (P.D.)
* Correspondence: james.mwase@ntnu.no

**Abstract:** This study investigates applying the principles of the long-discontinued Pedersen process as a possible route for producing metallurgical grade alumina from low-grade and secondary feed materials. The investigation focused on the hydrometallurgical steps in the process, namely leaching, desilication, and precipitation, and adapting it to valorize bauxite residue. The test material used was a calcium–aluminate slag made by the smelting-reduction of a mixture of bauxite residue (dewatered red mud) and a calcium-rich bauxite beneficiation by-product. Samples of the slag were leached in a 1 L jacketed glass reactor with $Na_2CO_3$ solution, varying $Na_2CO_3$ concentration and leaching time. Additionally, different approaches to leaching involving mechanical treatment of the leached slag and re-leaching using either fresh or recycled solution were also explored. The desilication step was carried out by treating the leachate solution with powdered CaO, varying the amounts of CaO used. Finally, the desilicated leach solution was sparged with a $CO_2$ gas mixture, after which the precipitate was allowed to age in the solution. The carbonation and aging temperatures and times were varied. As much as 67% of the Al was leached from the slag. The desilication process successfully removed 88% of the Si. The precipitation process produced a product composed mostly of bayerite $[Al(OH)_3]$, but some tests had considerable amounts of the unwanted phase dawsonite $[NaAlCO_3(OH)_2]$. The results indicated that the highest Al recovery was obtained using low concentrations of $Na_2CO_3$ solutions, and aluminum tri hydroxide is formed from these solutions at low temperatures at a fast rate compared to higher solution concentrations and temperatures.

**Keywords:** leaching; desilication; precipitation; alumina; Pedersen process





## 1. Introduction

The Bayer process has long been the premier, if not the exclusive, method for producing metallurgical grade alumina. Alumina is mainly used in the production of aluminum metal through high-temperature electrolysis and must be of high purity. The other application of alumina is making metal matrix composites [1]. However, the use of this process is limited to high-grade bauxite ores and is accompanied by the production of red mud. Red mud is a highly voluminous, highly alkaline, and hazardous solid waste product containing high amounts of iron, un-leached aluminum (15–25%), aluminum precipitated during solid/liquid separation, titanium, and rare earth elements [2]. A need for more environmentally friendly and sustainable methods to process mostly low-grade bauxite ores is the chief motivator behind the ENSUREAL initiative to investigate a potential revival of the Pedersen process. From 1928 to 1969, the process was successfully used to produce alumina at a 17,000 tpa plant situated in Høyanger, Norway [3]. In addition, the plant produced 4000 tpa of pig iron. It was eventually closed due to the economic strain of the less efficient smelter technology of the time [4]. This prohibited it from competing with the Bayer process in treating high-grade bauxite ores. However, improvements in smelter



technology, the need for more sustainable processes in the mining industry, and the need to address the red mud waste product from the Bayer process have renewed interest in this process. The process addresses all these matters while potentially being a zero-waste process producing by-products that can be used in other industries. Further, the Pedersen process has the potential to be used to remediate the red mud problem resulting from processing high-grade bauxite ores through the Bayer process.

The Pedersen process is a combined pyro- and hydro-metallurgical process [5–8], as illustrated in Figure 1. It can be conveniently described in three stages. First is the smelting of the bauxite ore at temperatures often exceeding 1500 °C. A reducing agent (carbon/coal) and flux (lime) are added to aid the process. This produces a slag that is chiefly composed of calcium–aluminate minerals and is easily separable from formed molten pig iron. The pig iron can potentially be processed to make cast iron. The most desired composition of the slag is to have Al in the form of mayenite ($Ca_{12}Al_{14}O_{33}$) and mono-calcium aluminate ($CaAl_2O_4$), which are the most leachable Al phases [9,10]. $SiO_2$ and $TiO_2$ are preferred in the form of $Ca_2SiO_4$ and $CaTiO_3$, respectively, which are difficult to solubilize and hence will not contaminate the final alumina product. In the second stage, the slags are leached with sodium carbonate solution to dissolve the aluminate minerals (reactions (1) and (2)) and leave behind a mostly calcium carbonate product referred to as grey mud. Unlike red mud, this by-product is environmentally benign and may be used as the principal feed material in the production of cement or fertilizer.

$$Ca_{12}Al_{14}O_{33} + 12Na_2CO_3 + 5H_2O \rightarrow 14NaAlO_2 + 12CaCO_3 + 10NaOH \qquad (1)$$

$$CaAl_2O_4 + 3Na_2CO_3 \rightarrow 2NaAlO_2 + CaCO_3 \qquad (2)$$

In the third stage, the leach liquor is then sparged with $CO_2$ gas to precipitate alumina tri-hydrate (reaction (3)).

$$2NaAlO_{2(aq)} + CO_{2(g)} + 3H_2O_{(l)} \rightarrow 2Al(OH)_{3(s)} + Na_2CO_{3(aq)} \qquad (3)$$

The precipitation process is very complex and starts with the hydrolysis of the sodium–aluminate ion (reaction (4)). It also involves the absorption of $CO_2$ gas by the sodium aluminate solution (reaction (5)), which leads to the neutralization of free hydroxyl ions by $H^+$ ions in solution. During this step, several physiochemical sub-processes, such as mass transfer through the gas-film, occur. This step creates a suitable pH for the hydrolysis (decomposition) of aluminate ions and precipitation of fine aluminum tri-hydroxide particles (reaction (6)). More details regarding the chemistry of this process are available [11–15].

$$NaAlO_{2(aq)} + 2H_2O_{(l)} = Na^+_{(aq)} + Al(OH)_4^-_{(aq)} \qquad (4)$$

$$CO_{2(aq)} + H_2O_{(l)} \leftrightarrow CO_3^{2-}_{(aq)} + 2H^+_{(aq)} \qquad (5)$$

$$Al(OH)_4^-_{(aq)} \leftrightarrow Al(OH)_{3(s)} + OH^-_{(aq)} \qquad (6)$$

The carbonation process is usually conducted in the temperature range of 60–90 °C for periods ranging from 5–10 h [9,16,17]. The quality and type of aluminum tri-hydroxide, as well as the morphology and particle size of the product, are influenced by the process parameters of pH, temperature, agitation speed, and duration of carbonation.

In the last stage, the aluminum tri-hydroxides are recovered from the slurry and dried. Then they undergo calcination, an endothermic dihydroxylation reaction, to convert them to metallurgical grade alumina according to reaction (7) [18].

$$Al(OH)_{3(s)} = Al_2O_{3(s)} + H_2O_{(g)} \qquad \Delta H^\circ_{(at\ 900\ °C)} = 62.5\ kJ/mol \qquad (7)$$

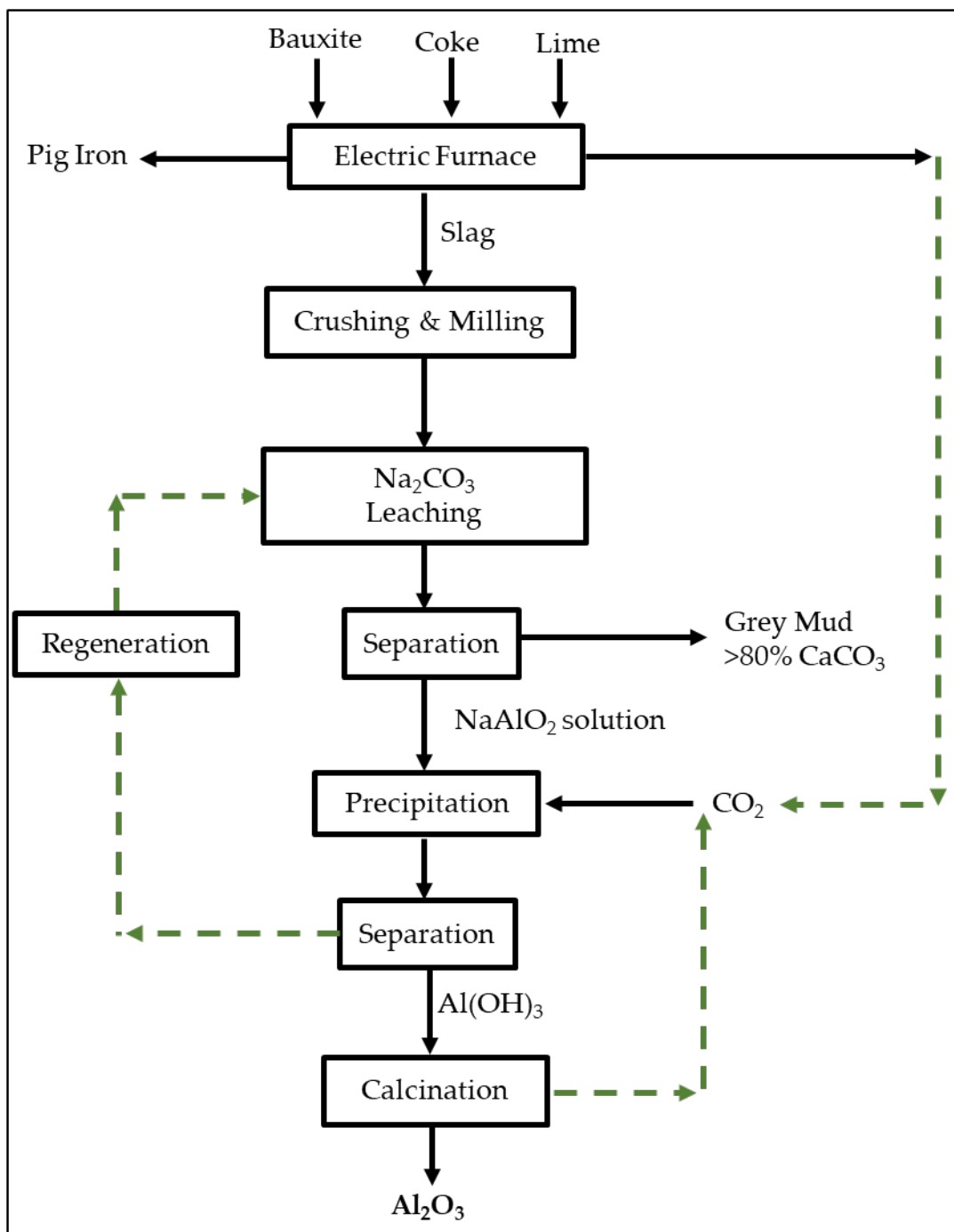

**Figure 1.** Mass flow diagram of the Pedersen process.

Detailed literature about the Pedersen process during the time it was in operation is not available due to confidentiality. The available literature is mostly short document patents and test reports [3,5,19] which have a very wide range and often conflicting process parameters. However, some studies have attempted to simulate the Pedersen process on ferruginous bauxite ores [9,20], and recently, the ENSUREAL initiative established to investigate the revival of the Pedersen process has conducted some studies whose results have provided data to better inform this study. The most notable outcomes from the ENSUREAL studies were the observations of the formation of a calcium-containing passivation layer around the slag grains, limiting the extraction of Al during leaching [21]. Further, leaching solutions with a $Na_2CO_3$ concentration higher than 100 g/L were observed to promote

dawsonite [NaAlCO$_3$(OH)$_2$] precipitation over the desired Al tri-hydroxides gibbsite and bayerite [22–24]. Additionally, it was noted that the precipitation process had to be a combination of carbonation and aging (allowing the precipitate to sit in solution for at least 24 h after carbonation) to convert boehmite to bayerite and gibbsite [9,23,24]. Furthermore, other studies [17,25] have reported that elevated temperatures exceeding 60 °C promote gibbsite formation.

The aim of this study is to determine the feasibility of using the Pedersen process to recover some value, chiefly in the form of aluminum, from secondary bauxite materials. These secondary materials are characterized by higher silicon and lower aluminum content than the usual bauxitic feed materials. The study will focus on evaluating (1) the extent of leaching of Al from a slag in which the Al is partially in the form of the mineral gehlenite (Ca$_2$Al$_2$SiO$_7$), (2) the effectiveness of using a desilication step to reduce the presence of Si in solution prior to precipitation to reduce the contamination in the alumina product, (3) the precipitation of the Al tri-hydrates from the leach liquors after desilication. It should be noted that gehlenite is the least desired form of Al in the slag as its formation is accompanied by Al loss. Slow cooling profile methods during smelting [26] have been shown to minimize the presence of this mineral. Due to its unique crystalline structure incorporating Si, gehlenite does not hydrolyze with water and has thus been known to be resistant to leaching even with aggressive chemical treatments [27–29].

## 2. Materials and Methods

### 2.1. Materials

The starting materials for the slag were bauxite residue (dewatered red mud), which is a waste residue from the Bayer process leaching step, and a calcium-rich bauxite ore by-product from the concentrator stage of the Bayer process. Both materials were supplied by Mytilineos, Greece. The chemistry of each material, determined by x-ray fluorescence (XRF), is detailed in Table 1.

**Table 1.** XRF quantitative analysis of bauxite residue and by-product (wt%).

| Elements | Bauxite Residue | By-Product | Elements | Bauxite Residue | By-Product |
|---|---|---|---|---|---|
| Al$_2$O$_3$ | 22.78 | 25.33 | Na$_2$O | 1.93 | - |
| CaO | 8.44 | 34.49 | MnO | 0.10 | - |
| TiO$_2$ | 5.88 | 1.24 | Cr$_2$O$_3$ | 0.26 | 0.01 |
| SiO$_2$ | 6.35 | 1.11 | P$_2$O$_5$ | 0.19 | - |
| Fe$_2$O$_3$ | 43.59 | 5.61 | NiO | 0.07 | - |
| MgO | 0.56 | - | S | 0.15 | 0.01 |
| V$_2$O$_5$ | 0.30 | - | LoI | 9.30 | 32.20 |

Prior to any test work, the materials, including the lime, were individually calcined for 3 h at a temperature of 900 °C to remove all moisture. The bauxite residue and the calcite-rich bauxite beneficiation by-product were mixed in a ratio of 3:2 by mass. The total mass of the mixture was 3 kg. To this mixture, lime (95.5% CaO) in a ratio of 21wt% of the mixture was added to obtain a proper chemical composition for the slag [26]. Half of the material was heated to 1650 °C in a cylindrical graphite crucible in an induction furnace (Inductotherm Europe, Droitwich Spa, UK). The remainder of the material was then added, resulting in a temperature drop. Some time was allowed for the temperature to reach 1650 °C once again, and it was held at that temperature for 40 min before the furnace was shut down and allowed to cool overnight. The resulting slag was separated from the pig iron produced and crushed and screened to produce a bulk sample with a particle size less than 75 μm. The material was then subjected to splitting using various size 2-way rifle splitters and an 8-way Retsch DR rotary splitter to produce sample sizes of 100 g for leaching test work. From the 100 g samples, smaller sub-samples were obtained using the quartering method for chemical analysis by XRF (Table 2). Mineral phase analysis via X-ray powder diffraction (XRD) (Figure 2) showed the slag contained Al in the form of mayenite

and gehlenite, with the rest of the slag being composed of perovskite ($CaTiO_3$) and larnite ($Ca_2SiO_4$). An effort was made to quantify the mineral phases by mass, but the results contradicted the spectrum (Figure 2). The intensity of the peaks shows that mayenite is the mineral phase that is most present and likely houses a significant portion of the Al.

**Table 2.** XRF quantitative analysis of slag (wt%).

| Al$_2$O$_3$ | CaO | SiO$_2$ | TiO$_2$ | Fe | Na$_2$O | SO$_3$ | MgO | ZrO$_2$ | MnO | SrO | Cr$_2$O$_3$ | P$_2$O$_5$ |
|---|---|---|---|---|---|---|---|---|---|---|---|---|
| 39.6 | 43.9 | 6.99 | 5.84 | 1.82 | 1.19 | 0.18 | 0.17 | 0.14 | 0.06 | 0.06 | 0.03 | <0.05 |

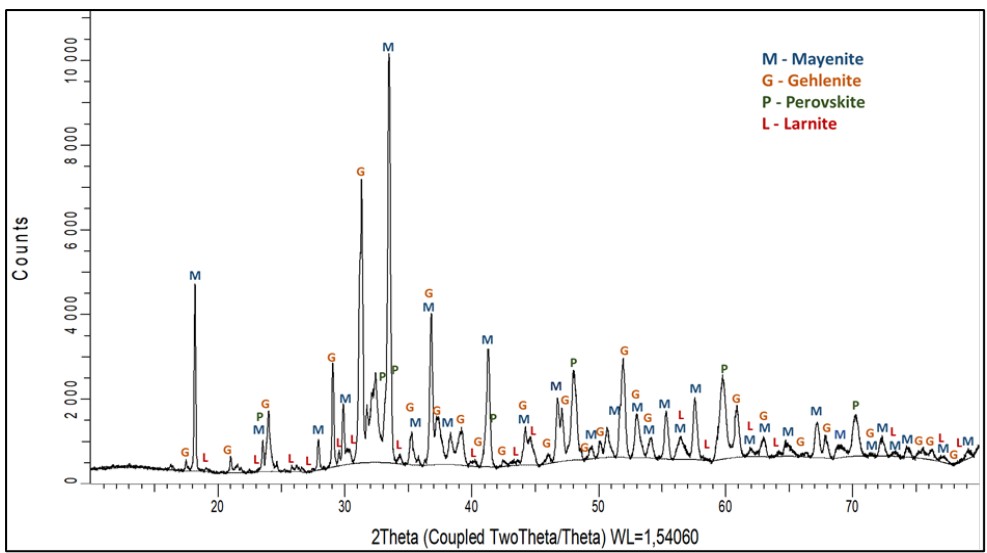

**Figure 2.** XRD spectrum of slag with the identified phases.

The microstructure of the slag was further characterized by electron microprobe analysis (EMPA) supported by a wavelength dispersive spectrometer (WDS). A sample of the slag was mounted onto polished resin blocks, and analyses included elemental mapping, imaging, and quantification. The extract (Figure 3) from the total elemental mapping for the slag shows it is rich in Al and Ca, with Ti and Si being the elements next in abundance. Element quantification performed at the six selected points (Figure 4) by percentage mass (Table 3) confirms the presence of the minerals identified by XRD analysis. This was done by comparing the masses with the chemical formulas of the minerals. The Al/Ca mass ratio is 1.27 and 1.48 for the stoichiometric compositions of mayenite and gehlenite, respectively. However, as both phases can have levels of silica dissolved [30], they have some deviation from the stoichiometric ratio. The Al/Ca ratio in points 1 and 2 suggested a higher presence of mayenite, while the Al/Ca ratio, along with the high Si content, at points 3 and 4 suggested the gehlenite in these parts was more than mayenite and possibly had the highest content of larnite among the six points selected. Points 5 and 6 had the highest amounts of Ti, showing higher amounts of perovskite.

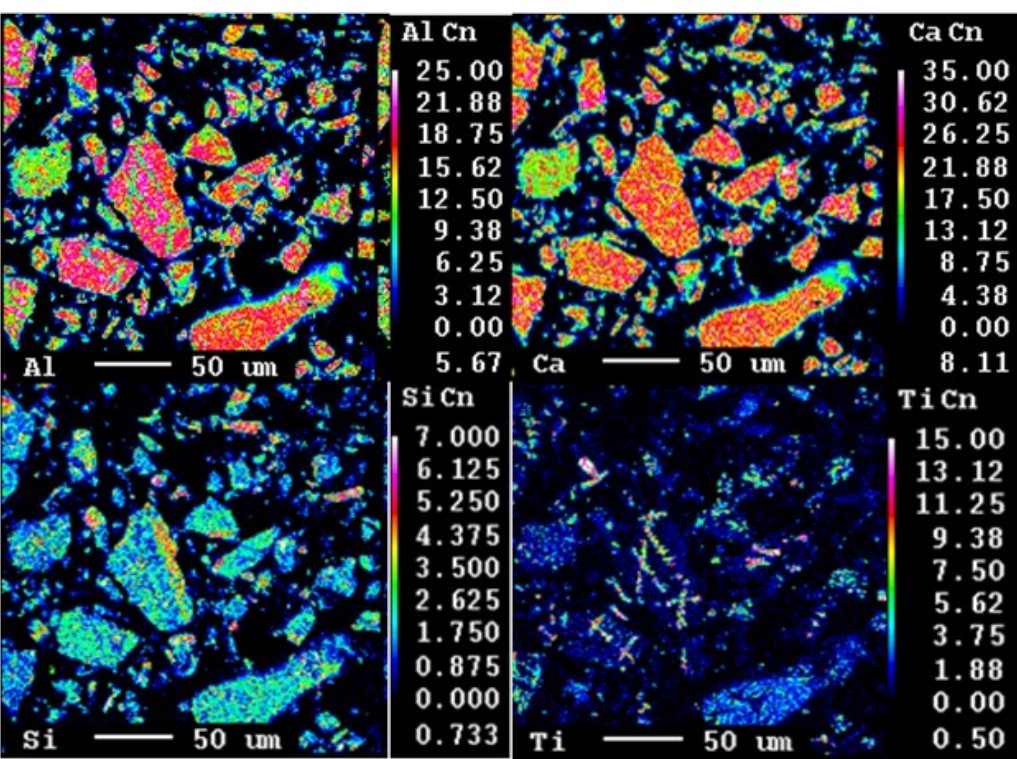

**Figure 3.** Elemental mapping of the slag using EMPA.

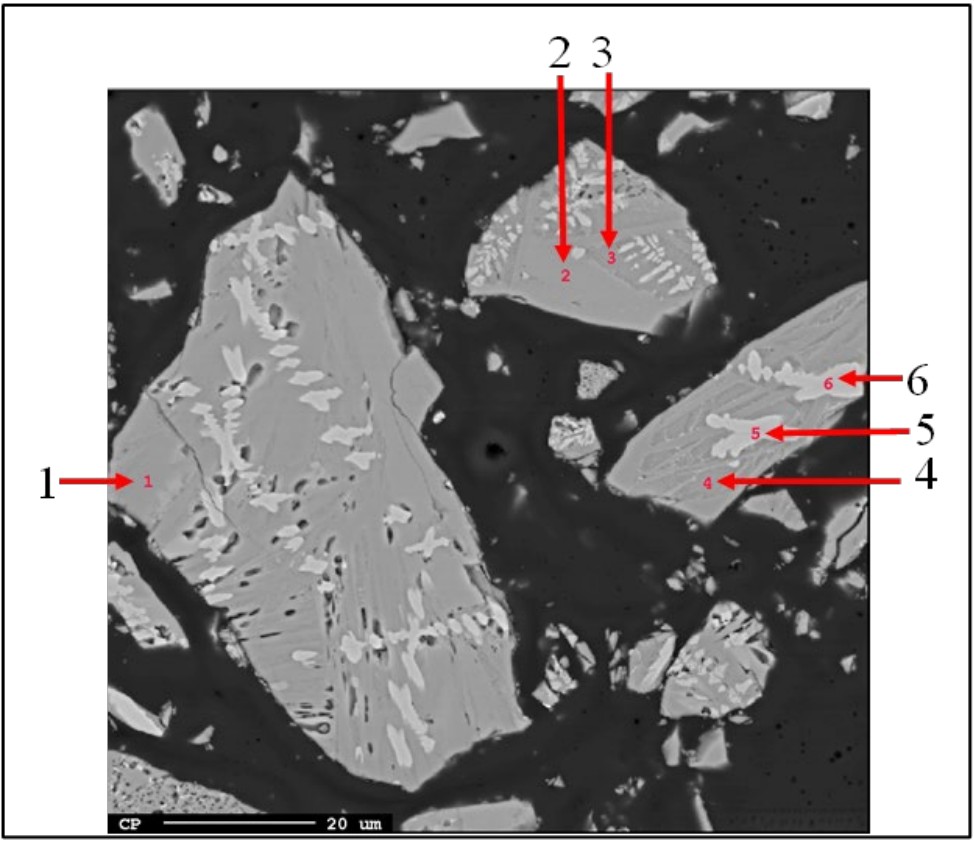

**Figure 4.** EPMA micrograph and the analyzed points for characterization. Points 1 and 2 (mayenite), points 3 and 4 (gehlenite), points 5 and 6 (perovskite).

**Table 3.** Percentage mass quantification of selected elements in the slag by EMPA.

| No. | $Al_2O_3$ | CaO | $SiO_2$ | $TiO_2$ | Al/Ca | Proposed Phase |
|-----|-----------|------|---------|---------|-------|----------------|
| 1 | 52.65 | 1.49 | 4.16 | 0.41 | 1.06 | Mayenite |
| 2 | 51.00 | 1.72 | 4.37 | 0.60 | 1.10 | Mayenite |
| 3 | 41.76 | 8.26 | 5.76 | 0.73 | 1.56 | Gehlenite (+larnite) |
| 4 | 40.48 | 8.63 | 5.61 | 1.36 | 1.62 | Gehlenite (+larnite) |
| 5 | 15.71 | 6.36 | 1.98 | 34.43 | - | Pervoskite |
| 6 | 16.42 | 6.88 | 1.97 | 35.16 | - | Perovskite |

*2.2. Methods*

A total of 6 test programs were conducted using a 1 L jacketed glass reactor. The same reactor was used for all three processes. After each individual test, the reactor was thoroughly cleaned in preparation for the next test.

2.2.1. Leaching Tests

Each test program commenced with a leaching test in which a 100 g sample of slag was leached with 1 L of $Na_2CO_3$ solution in the reactor. The leaching solution was prepared by dissolving a specified amount of analytical grade $Na_2CO_3$ in 1 L of deionized water. The resulting slurry was heated to the desired temperature and held at that temperature for the desired duration. The temperature was controlled by circulating a heated silicon oil through the reactor jacket and was monitored using a temperature probe placed in the reactor vessel. The slurry was agitated using an overhead stirrer with a paddle impeller. The temperature probe was connected to the agitator, which displayed the temperature in addition to the agitation speed. All tests were conducted at 90 °C with an agitation rate of 500 rpm. The volume of the reactor was maintained using a water-cooled condenser. The tests varied in terms of different concentrations of $Na_2CO_3$ solution used and leaching durations. Additionally, some NaOH was added to test no. 2 to determine if its addition would result in less Si dissolved [3,19]. It was worth investigating this, although previous studies have shown that the addition of NaOH reduced Al extraction [21]. Further, for some tests, the leached sample residue was recovered, dried, and subjected to physical treatment using a pestle and mortar. The force provided by the pestle and mortar was not expected to grind or break particles but perhaps help remove the passivation layer of $CaCO_3$ over the particles that are formed via reaction (1) [21]. The sample was then re-leached with recycled solution or with fresh solution. All tests are detailed in Table 4. Depending on the duration of the test, solution samples were taken at intervals of 15 and 30 min and vacuum filtered using a 0.22 μm filter for inductively coupled plasma mass spectrometry (ICP-MS) analysis. At the end of the tests, the slurries were vacuum filtered with Whatman general-purpose filter paper to separate the solid residue (grey mud) and leachate solution. The grey mud was dried overnight at 100 °C and analyzed via XRD to determine the mineral phases and XRF for chemical composition for a material balance. The leachate solution was allowed to cool overnight for the desilication step.

**Table 4.** Schedule of leaching tests using 100 g slag, 1 l solution, 90 °C leaching temperature, and 500 rpm agitation rate.

| Test No. | First Stage Leach | Intermediate Step | Second Stage Leach |
|---|---|---|---|
| 1 | 60 g/L $Na_2CO_3$ (90 min) | Physical treatment with pestle and mortar | Recycled solution from 1st Stage Leach (90 min) |
| 2 | 60 g/L $Na_2CO_3$, 4 g NaOH (90 min) | N/A | N/A |
| 3 | 60 g/L $Na_2CO_3$ (180 min) | N/A | N/A |
| 4 | 100 g/L $Na_2CO_3$ (90 min) | Physical treatment with pestle and mortar | Recycled solution from 1st Stage Leach (90 min) |
| 5 | 40 g/L $Na_2CO_3$ (90 min) | Physical treatment with pestle and mortar | Leach with fresh solution 40 g/L $Na_2CO_3$ (90 min) |
| 6 | 80 g/L $Na_2CO_3$ (180 min) | N/A | N/A |

2.2.2. Desilication Tests

The desilication method, adapted from [9], was conducted on the solutions from all tests, except for test no 5. ICP-MS analysis showed the Al in solution in test no. 5 was less than in the other tests. A sample of the solution was taken at the start and end of the test and vacuum filtered with a 0.22 μm membrane for ICP-MS analysis of Si. It should be noted that the volumes of the solutions in these tests were less than 1l. After filtration, the grey mud from the leaching tests retained some solution, and hence the volume of solution recovered from the experiments ranged from 800 mL to 870 mL. All the tests were conducted by adding the leachate solution to powdered CaO in the glass reactor. A total of 14 g of CaO was used for the solutions from tests no. 1 and 2, while 28 g was used for the solutions from tests no. 3, 4, and 6. In each test, the mixture was heated to 70 °C and agitated at 400 rpm for 60 min. At the end of each test, the solution was vacuum filtered with general-purpose filter paper. The solution was allowed to cool overnight for the precipitation tests. The solid residue was dried at 60 °C for 2 days and analyzed via XRD and XRF.

2.2.3. Precipitation Tests

The precipitation tests were conducted on all the desilicated solutions as per Table 5. Like the leaching tests, the filter cake from the desilication tests retained some solution, and hence the volumes recovered were less than the starting amount. The solution volumes for the precipitation tests ranged from 700 mL to 800 mL. In all the tests, the desilicated solutions were sparged with a gas mixture equal parts Ar and $CO_2$. The Pedersen process used an off-gas from the smelter operation in the precipitation process [3]. The exact composition of the gas is not known, but it has been established that $CO_2$ was the only active component and was present in a potentially low partial presence. As a first attempt to replicate the process, the current conditions have been chosen as a starting point, absent data on the exact conditions of the Pedersen process. For every test, the gas flow rate was set to 1.5 slpm (standard liters per min), and the solution was agitated at a rate of 200 rpm. The solutions were sparged for a certain interval, and the mixture (liquid and precipitate) was allowed to age for different periods of time. At the end of the test, the mixture was vacuum filtered with general-purpose filter paper to recover the precipitate. The precipitate was washed with deionized water and dried at 60 °C for 2 days. Samples of the precipitate were taken for XRD, XRF, and particle size distribution (PSD) analyses. Solution samples were taken before and after the test and vacuum filtered with a 0.22 μm membrane for ICP-MS analysis of the Al.

**Table 5.** Schedule of precipitation experiments.

| Test No. | Solution from Leach Test No. | Temp. (°C) | $P_{CO_2}$ (%) | Rate of Agitation (rpm) | Duration (h) | Aging (days) |
|---|---|---|---|---|---|---|
| 1 | 1 | 25 | 50 | 200 | 0.5 | 3 |
| 2 | 2 | 25 | 50 | 200 | 0.5 | 3 |
| 3 | 3 | 50 | 50 | 200 | 1 | 1 |
| 4 | 4 | 75 | 50 | 200 | 2 | 1 |
| 5 | 6 | 25 | 50 | 200 | 2 | 1 |

All XRF analysis was conducted by Degerfors Laboratorium AB, Sweden, while all XRD, ICP-MS, EMPA, and PSD analyses were conducted at NTNU and SINTEF labs.

XRD analysis was conducted using Bruker D8 A25 DaVinci™ X-ray Diffractometer with CuKα radiation, 10 to 75 degrees. diffraction angle, and scanning speed 0.01 degrees. EVA and TOPAS software were used to interpret data against the Crystallography Open Database.

Samples for ICP-MS from leaching, desilication, and precipitation tests were diluted in 15 mL Millipore water with 7 drops of 0.1 M nitric acid in preparation for analysis.

EMPA was conducted on samples in polished resin blocks using a JEOL JXA-8500F Electron Probe Micro Analyzer (EPMA) with Wavelength Dispersive X-ray Spectrometer with acceleration voltage 15.0 kV.

PSD analysis was conducted by Sintef laboratories using a Coulter LS 230 instrument and solvent 0.05% Calgon.

## 3. Results and Discussion

### 3.1. Leaching

3.1.1. Leaching Behaviour of the Slag

Table 6 shows the percentage extractions of Al achieved as calculated from ICP-MS analysis of solution samples (Equation (1)) and from a material balance performed by using XRF analysis on the slag and the resulting grey mud residue (Equation (2)). For all the tests except test no. 5, the results determined by the 2 methods are comparable, with less than a 10% difference between the figures. In the case of test no. 5, it was not possible to re-run the analysis; thus, the discussion will focus on the other five tests. Despite varying certain process conditions and using different approaches to leaching, the extractions of Al were similar across the six tests. Going into the test program, it was not known to what extent the gehlenite would limit the extent of Al leached, given that the amount of gehlenite in the slag could not be quantified. The extractions achieved suggest that the gehlenite housed about 30–35% of the Al, provided that the gehlenite does not leach. Further, the Al extractions achieved are comparable to those achieved in lab-scale and pilot tests conducted on slags that are mostly mayenitic with little or no gehlenite [31]. This study [31] used agitated leaching with no intermediate grinding; therefore, the extractions in their studies may be due to passivation brought on by factors such as high $Na_2CO_3$ concentration (120 g/L). In this study, however, the extent of leaching is attributed to the presence of gehlenite, as will be discussed later.

$$\text{Al recovery (\%)} = \frac{\text{mass of Al in leachate}}{\text{mass of Al in slag}} \times 100 \qquad (8)$$

$$\text{Al recovery (\%)} = \left(1 - \frac{\text{mass of Al in grey mud}}{\text{mass of Al in slag sample}}\right) \times 100 \qquad (9)$$

**Table 6.** Al percentage extraction determined by ICP-MS analysis of solution samples and material balance via XRF analysis.

| Test No. | 1 | 2 | 3 | 4 | 5 | 6 |
|---|---|---|---|---|---|---|
| ICP-MS | 63.53 | 61.20 | 57.13 | 58.16 | 42.58 | 53.03 |
| XRF | 67.12 | 64.95 | 64.17 | 64.60 | 68.77 | 58.25 |

3.1.2. Solution Concentration

The leach curves for tests no. 1 and 4 can be seen in Figure 5. The two tests were identical, except they used different solution concentrations for leaching. In test no. 1, the intermediate grinding step and re-leaching with recycled solution did not result in more Al being extracted. In comparison, the leach curves for test no. 4 show that the identical treatment resulted in an increased amount of Al leached in the second stage leach. It is noted that in the first stage leach of test no. 4, less Al was extracted, and the graph more or less leveled off. This could be because passivation via the $CaCO_3$ formation occurred faster in this case due to a higher solution concentration used. However, the intermediate mechanical treatment, resulting in the temporary removal of the passivation layer and possible deagglomeration, and contacting again with the solution gives the maximum recovery experienced by the two solutions. Comparing the highest recoveries after the second stage leaching trials, we may conclude that the passivation that occurred for test no. 4 was due to the higher solution concentration used.

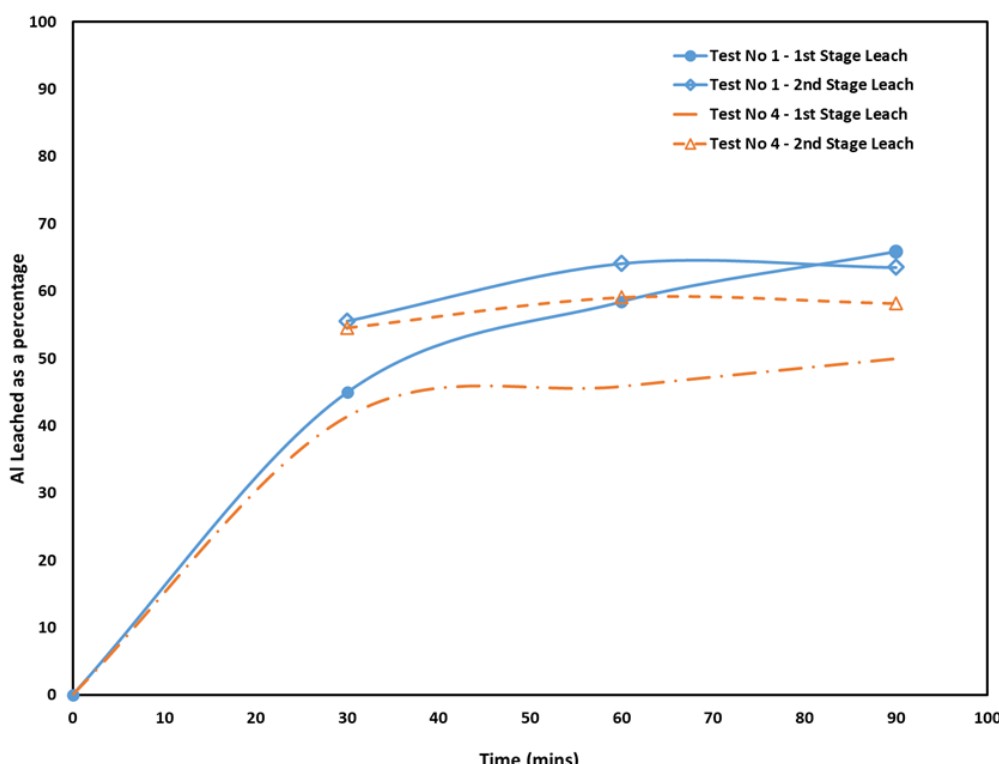

**Figure 5.** Leach curves from test no. 1 (60 g/L $Na_2CO_3$) and test no. 4 (100 g/L $Na_2CO_3$).

Figure 6 shows the Al recoveries achieved with respect to the solution concentration. It indicates that, in general, the increase in solution concentration for a given leaching time yields a lower Al recovery. It is worth mentioning that the trendlines are not showing linear regression; however, they are added to indicate the average changes of Al recovery. Moreover, this figure shows that the leaching recovery change with time is larger when lower solution concentrations are used. For the highest applied solution concentration of

100 g/L, it is seen that the Al recovery has been increased minimally, and this indicates that the passivation by $CaCO_3$ formation occurs faster.

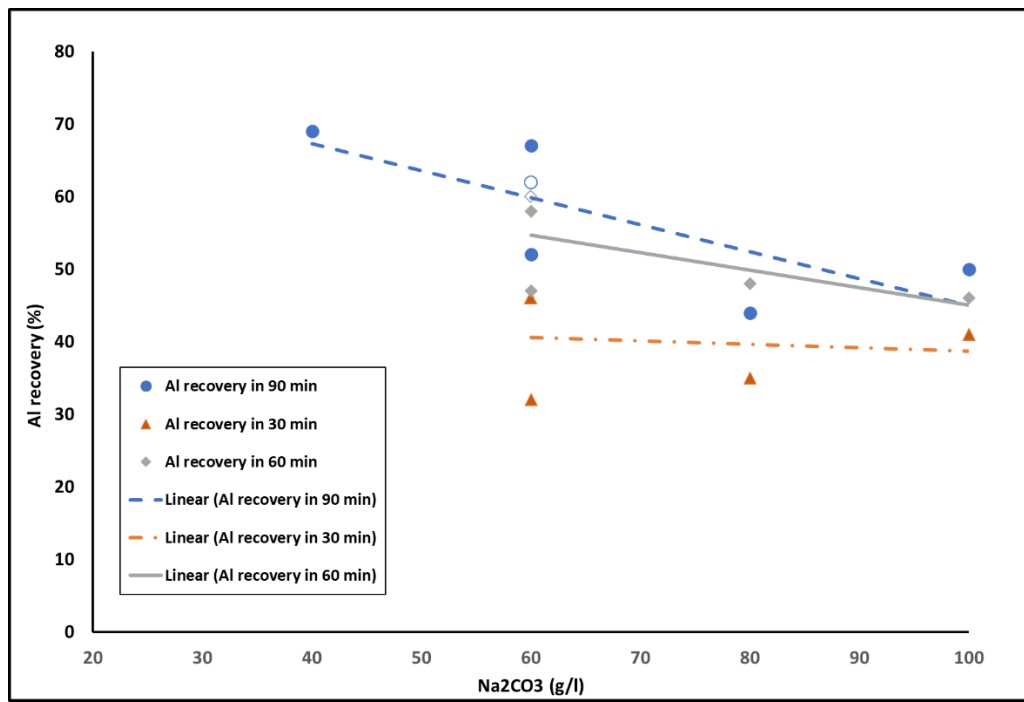

**Figure 6.** Comparison of Al recovery at various time points from leach curves of tests no. 1, 2, 3, 4, and 6.

### 3.1.3. NaOH Addition

From Figure 7, it appears that the presence of NaOH periodically hindered Al extraction, although ultimately, the final amount in 90 min leaching was comparable to the amounts extracted in the tests without NaOH. Further ICP-MS analysis showed the amount of Si dissolved in test no. 2 was 0.15 g/L which was comparable to tests no. 1 and 3, which had concentrations of 0.2 and 0.15 g/L, respectively. Further still, tests no. 4 and 6, which had the highest concentrations of $Na_2CO_3$, each had Si concentrations of 0.33 g/L. Therefore, the presence of NaOH did not reduce Si dissolution, and $Na_2CO_3$ concentration had more influence on Si dissolution than NaOH. As a result, the use of NaOH in further tests was discontinued. The added NaOH is low compared to the stoichiometric amount of NaOH that is produced according to reaction (1) and compared to the amounts added by Azof et al. [21], which resulted in lower Al leached. Azof et al. [21] added NaOH, resulting in a $Na_2O$ concentration of 60 g/L. Therefore, it is unlikely that the amount added had any impact on the leaching. It is not known definitively why the curve for test no. 2 is non-linear, as the samples were tested with two different labs and the pattern remained consistent. The outlier points could be interference caused by the additional amount of Na in the ICP-MS analysis. Further, it may have been precipitation of Al at those specific times in the reaction, or it might have occurred in the sample bottles after the samples were taken.

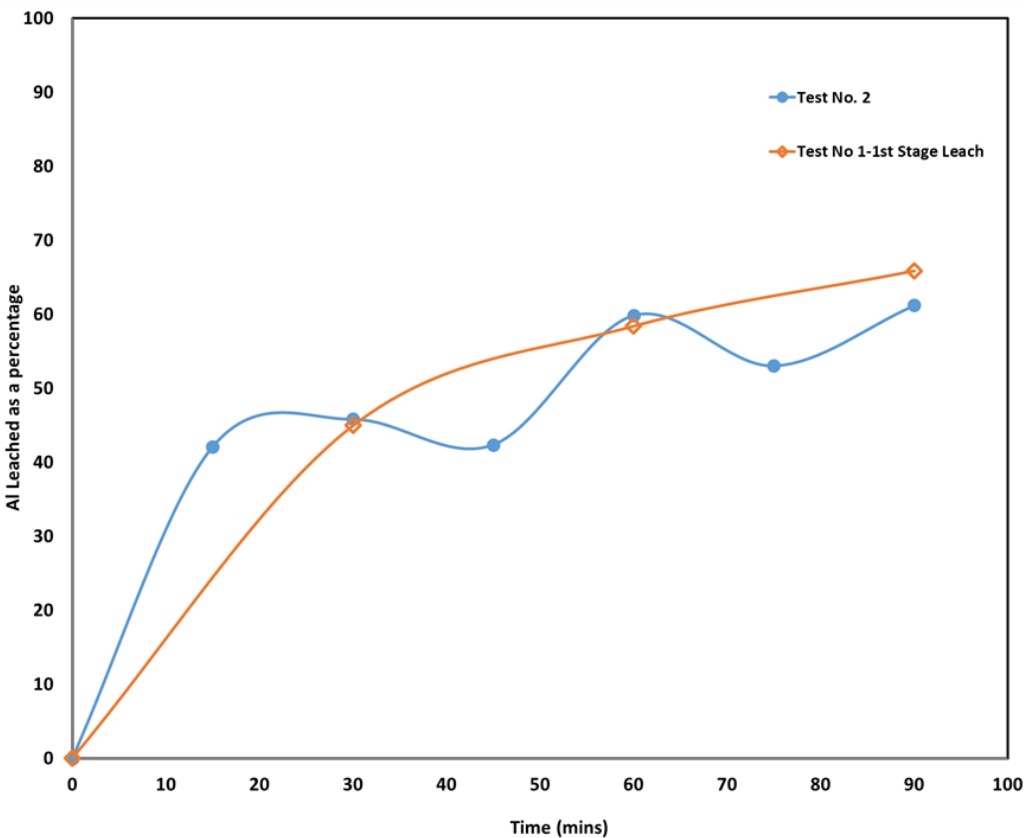

**Figure 7.** Test no. 2 leaching of 100 g of slag in the presence of 4 g NaOH.

### 3.1.4. Leaching Duration and Solution Concentration

Test no. 3 was carried out regarding the result from the first stage leach of test no. 1. The curve for test no. 1 has not leveled off in 90 min leaching and suggests that perhaps more Al would be extracted. Therefore, test no. 3 was conducted under identical conditions with the exception that the leaching time was extended to 180 min. However, this did not result in higher Al extraction, as seen from the leach curve in Figure 8. After 90 min, the curve begins to level off, but the final extraction is comparable to all the other tests. The difference in Al recovery may be due to the sample leached in test no. 3 possibly having had less mayenite and more gehlenite than the sample leached in test no. 1. Further, larger particle sizes and possible agglomeration in the sample for test no. 3 could be another cause, as this is a phenomenon that is corroborated by literature [32]. The final test to be discussed, test no. 6 (Figure 9), was also motivated by the result of the first stage leach from test no. 1. In this case, it was allowed to run for an extended time, similar to test no. 3 except a higher concentration (80 g/L) of $Na_2CO_3$ was used. The leach curve shows that at the 90 min mark, the Al extraction dips, but further than that, it increases, and finally, at the 180 min, it dips slightly, but again, the final extraction is comparable to all the other tests. This point, however, could be an outlier, and if removed, the curve appears smoother. In all experiments, the rate of leaching is initially fast and then slows down when 40–50% of Al is leached. Then, the Al recovery is at a slow rate and almost leveled off after 90 min and longer times.

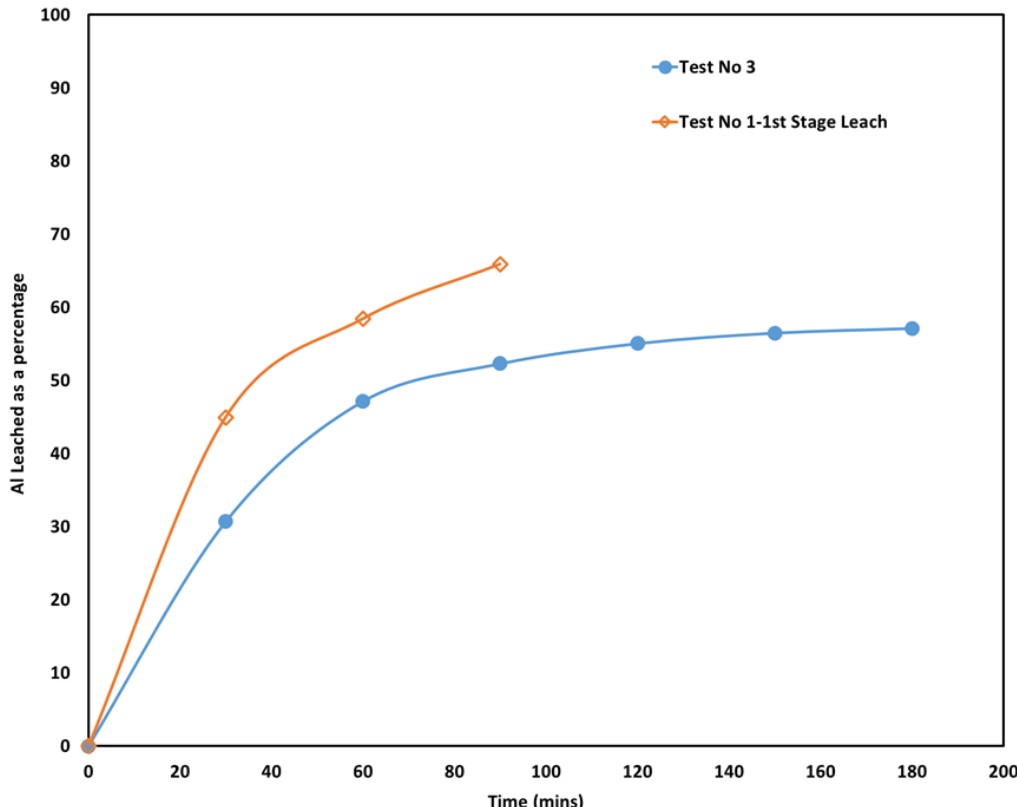

**Figure 8.** Test no. 3 leaching of 100 g of slag 1L of 60 g/L $Na_2CO_3$.

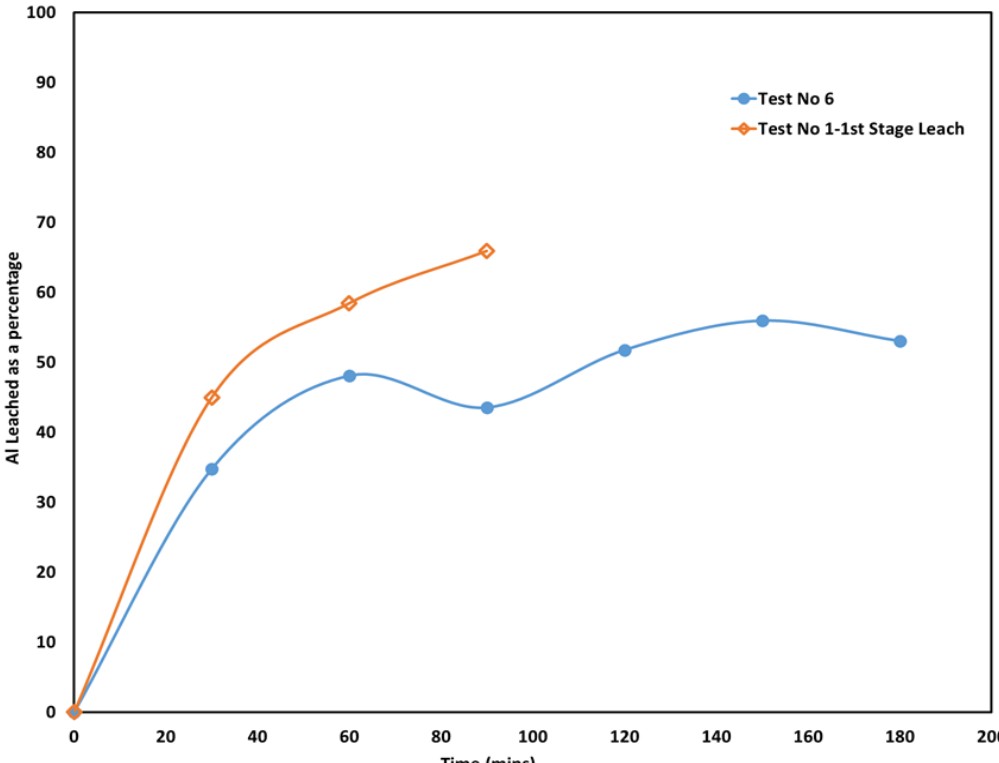

**Figure 9.** Test no. 6 leaching of 100 g of slag 1L of 80 g/L $Na_2CO_3$.

### 3.1.5. Grey Mud Analysis

Table 7 shows the mineral phase analysis of the grey mud determined by quantitative XRD analysis. This quantification was deemed reliable when compared to the peaks in the XRD spectrum (Figure 10). The analysis shows that the mayenite was mostly or completely leached out in all samples, and there is a considerable amount of calcite that has formed. The weight% of calcite formed, considering reaction (1), would suggest that mayenite in the slag was probably 60% by mass. The remaining Al sits in the gehlenite phase and is un-leachable. It is also interesting to note the presence of Na-Zeolite and Bayerite as minor phases in the grey mud. The former, a sodalite-type material, is a typical desilication product in aluminate solutions, and it is also observed in the pre-desilication of bauxite in the Bayer Process [33]. The latter is an aluminum trihydrate phase which is suggestive of saturation conditions in the aluminate solution, leading to its precipitation. Both these phases highlight the complexity of the chemical actions taking place in parallel to the main hydrometallurgical process of leaching.

**Table 7.** XRD semi-quantitative analysis of the grey mud samples (wt%).

| Test No. | 1 | 2 | 3 | 4 | 5 | 6 |
|---|---|---|---|---|---|---|
| Mineral Phase | | | | | | |
| Calcite ($CaCO_3$) | 56 | 46 | 54 | 66 | 59 | 56 |
| Gehlenite | 20 | 24 | 21 | 7 | 16 | 17 |
| Perovskite | 15 | 17 | 14 | 14 | 17 | 16 |
| Larnite | 7 | 12 | 10 | 5 | 8 | 9 |
| Mayenite | 1 | 1 | 0 | 0 | 0 | 0 |
| Zeolite A ($NaAlSiO_4$) | 0.5 | 0 | 0.5 | 8 | 0 | 2 |
| Bayerite | 0.5 | 0 | 0 | 0 | 0 | 0 |

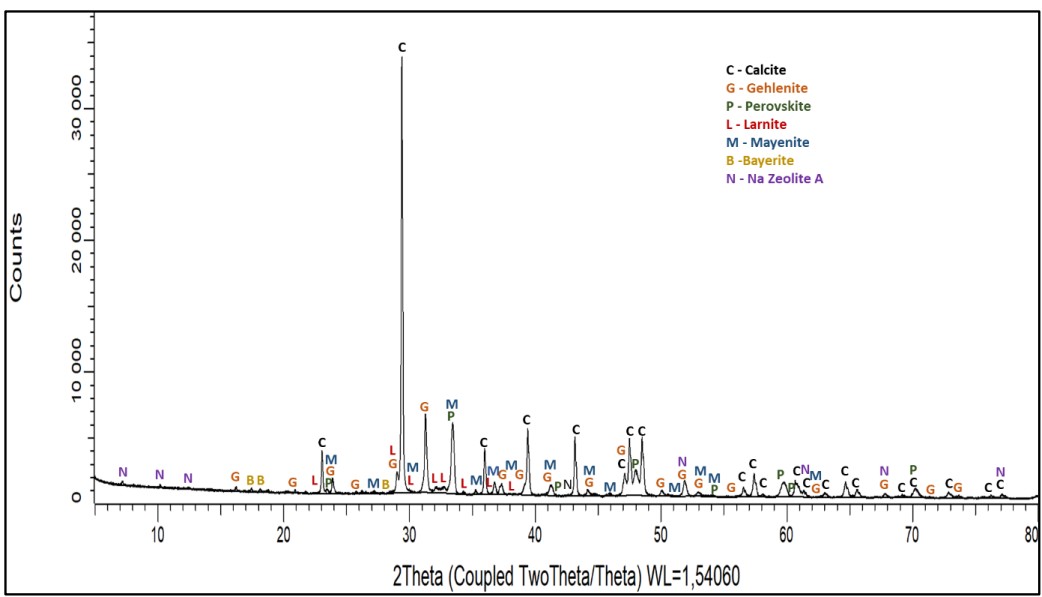

**Figure 10.** A typical XRD spectrum from grey mud sample from leaching test no. 1.

EMPA identical to the one conducted on the slag sample was conducted on grey mud samples from tests no. 1 and 3. Figure 11 shows that, while the sample is still Ca-rich, the Al has reduced somewhat in comparison to the slag (Figure 3). Obviously, Al concentration at the surface and close to the surface of particles is lower than the core, while Ca concentration at the surface is higher. Comparing the image of the grey mud in Figure 12 with that of the slag (Figure 4), the slag particles have smooth edges around them,

while the edges of leached samples are irregular and porous. The layers around the leached particles (from both tests) are visible, and the quantification analysis suggests they are mostly composed of calcium carbonate with some larnite. However, this layer is possibly the reason why there are minimal amounts of un-leached mayenite in tests no. 1 and 2. The circled points of the particle in Figure 12A are the gehlenite phase which is not passivated but remains un-leached. The color of this area is identical to the one identified as gehlenite from Figure 12B. The quantification analysis at the marked points in Figure 12B gives an idea of the mineral deportment in a particle that appears to be mostly calcite ($CaCO_3$). The high amounts of Ca and Si in areas 1 and 2 suggest they are mostly calcite with some larnite. Higher amounts of Al and Ca with less Si suggest 3 and 4 are gehlenite with possibly some of the un-leached mayenite, while the high amounts of Ti suggest that areas 5 and 6 have the highest concentration of perovskite.

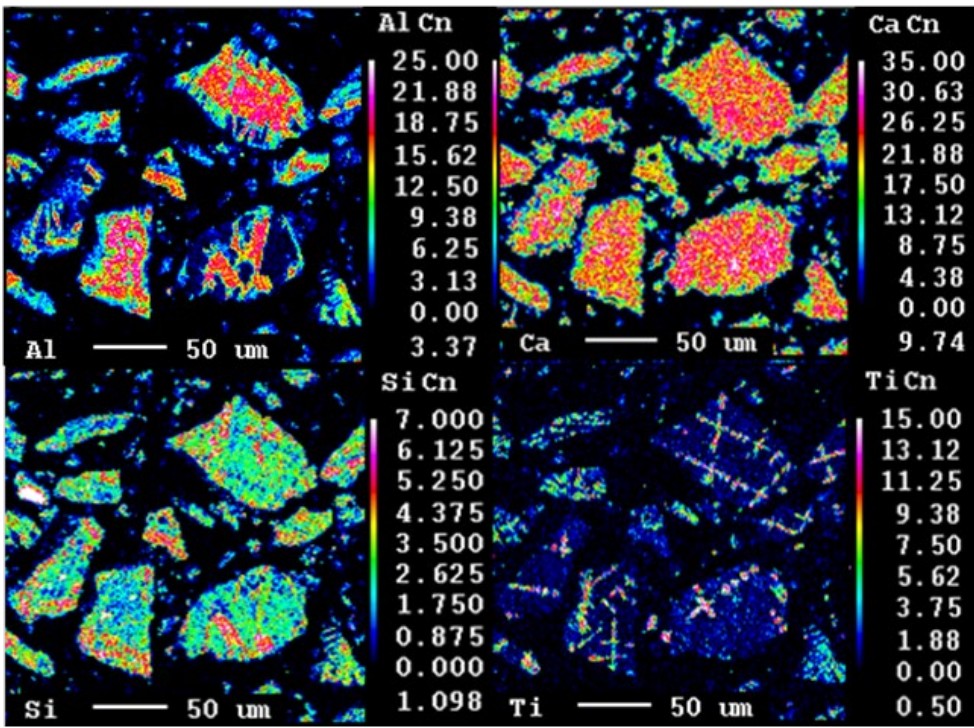

**Figure 11.** Elemental mapping of grey mud from test no. 1.

Regarding the size of the slag particles, mostly below 75 μm (as seen in Figure 4), most of the particles have a size close to or smaller than the completely leached particles in Figure 12B. Therefore, we may conclude that most of the particles have been leached regarding the digestion of mayenite. This may be the reason that we obtained the highest Al recovery in leaching at 68%, and significantly higher Al recovery is not expected, as gehlenite is not leachable. In other words, if the Al recovery is defined as the Al extraction from the leachable phase mayenite, we have obtained more than 90% Al recovery from mayenite in tests no. 1, 2, and 5.

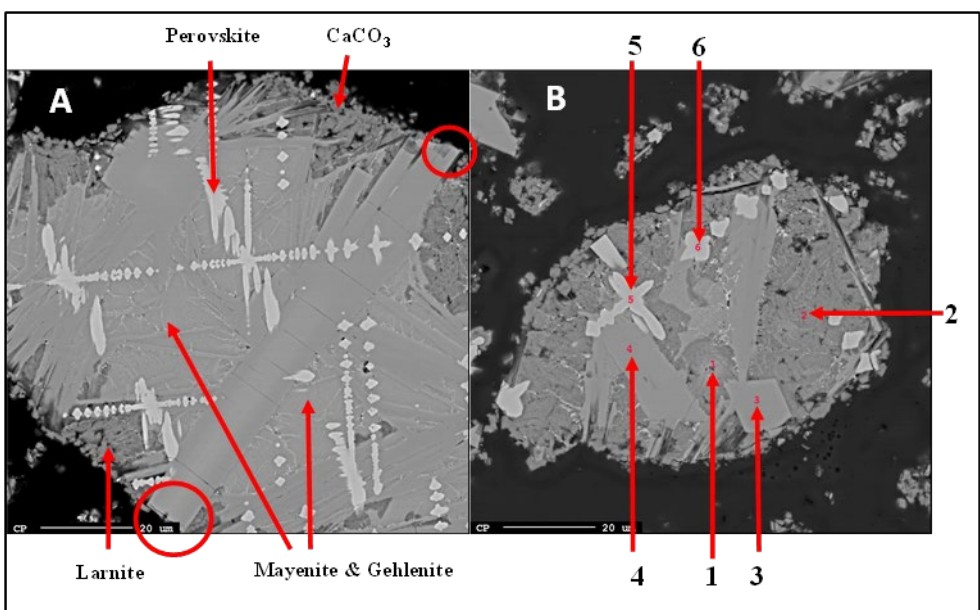

**Figure 12.** (**A**) Image from grey mud sample test no. 1. (**B**) Elemental quantification on grey mud sample test no. 3. Points 1 and 2 (mayenite), points 3 and 4 (gehlenite), points 5 and 6 (perovskite).

*3.2. Desilication*

The results (Table 8) show that, overall, the desilication step does reduce the concentration of Si in solutions. However, there is some inconsistency in the results. The use of an increased amount of CaO (28 g) results in a much higher Si removal for tests no. 3 and 6 than when 14 g was used in tests no. 1 and 2 but not 4. Test no. 4 had the highest amount of Na, and although the chemistry cannot be explained, this may have caused the low Si removal despite using a high amount of CaO. Additionally, there was co-precipitation of Al in all the experiments, as seen from the mineral phases present (Table 9). A sample of the XRD spectrum from test no. 6 can be seen in Figure 13. The existence of hydrogarnite in the desilication residue (tests no. 3, 4, 5) indicates the simultaneous precipitation of both Si and Al is taking place, yielding the removal of two moles of Al and three moles of Si from the solution. Higher Al loss from the solution is, however, related to the crystallization of calcium monocarboaluminate.

**Table 8.** Concentration of Si and Al in solution measured by ICP-MS and precentage removal of Si and Al from solution.

| Test No. | 1 * | 2 * | 3 ** | 4 ** | 6 ** |
|---|---|---|---|---|---|
| Start (g/L) | 0.2 | 0.15 | 0.15 | 0.31 | 0.32 |
| End (g/L) | 0.15 | 0.11 | 0.02 | 0.23 | 0.04 |
| Silicon removal (%) | 25 | 27 | 87 | 28 | 88 |
| Start (g/L) | 15.07 | 13.17 | 10.04 | 23.10 | 22.23 |
| End (g/L) | 12.04 | 9.88 | 5.38 | 20.90 | 18.30 |
| Aluminum removal (%) | 26.64 | 30.09 | 55.08 | 18.46 | 26.43 |

* 14 g CaO added, ** 28 g CaO added.

**Table 9.** XRD semi-quantitative analysis of phases in the desilication filter cake (wt%).

| Test No. | 1 | 2 | 3 | 4 | 6 |
|---|---|---|---|---|---|
| Mineral Phase | | | | | |
| Calcium monocarboaluminate [$Ca_4Al_2(OH)_{12}CO_2.5H_2O$] | 70 | 58 | 67 | 56 | 44 |
| Calcite ($CaCO_3$) | 21 | 33 | 11 | 36 | 31 |
| Portlandite [$Ca(OH)_2$] | 9 | 9 | 10 | 6 | 8 |
| Hydrogarnite $Ca_3Al_2(SiO_4)_{3-x}(OH)_{4x}$ [x = 1.5–3] | 0 | 0 | 12 | 1 | 18 |

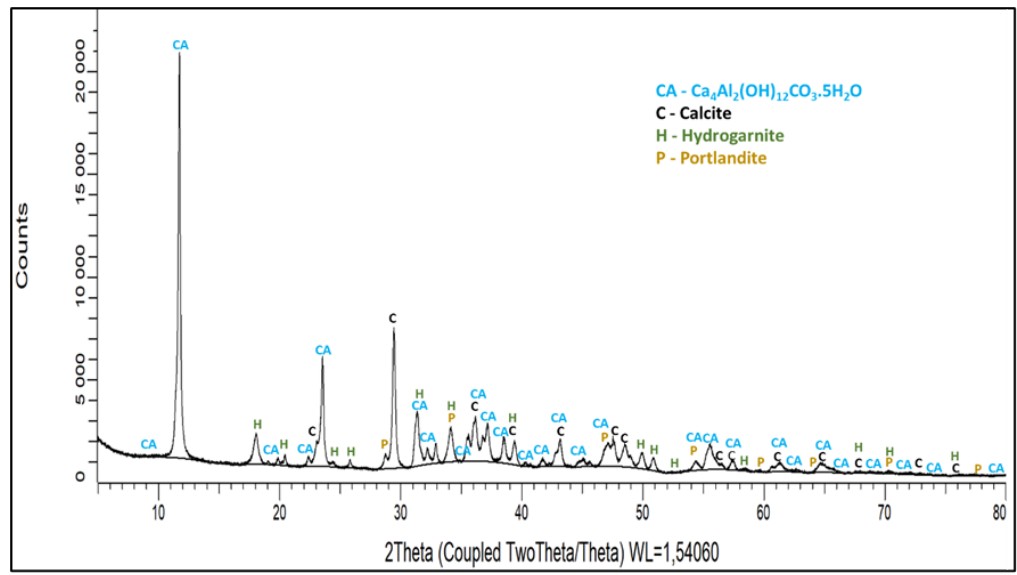

**Figure 13.** XRD spectrum for desilication residue from test no. 6.

The Eh-pH diagram for the Si-water system was evaluated by HSC Chemistry software, and it was found that at high pH and different concentrations of silica, the Si-containing ions were $H_2SiO_4{}^{2-}$ and $H_3SiO_4{}^{-}$. The change in silica concentration and temperature changes were studied, and it was found that the $H_2SiO_4{}^{2-}$ is more stable than $H_3SiO_4{}^{-}$ at lower concentrations and lower pH. Moreover, temperature change between 25 and 90 °C does not affect their stabilities significantly. Proposed chemical reactions that yield hydrogarnite in the system via the interaction of $H_2SiO_4{}^{2-}$ and $H_3SiO_4{}^{-}$ with CaO or $Ca^{2+}$ could not be balanced. However, Nikolaychuk [34] proposed equilibria information about the Si-water system, similar to conditions in this study, which had more species than the HSC database. From the revised pourbaix diagrams [34], the species of Si that is most likely present in the study's conditions was $H_7SiO_6{}^{-}$. Hence, reaction (8) can be proposed as a possible mechanism for the desilication process.

$$3Ca^{2+} + 2AlO_2{}^{-} + (3-x)H_7SiO_6{}^{-} + (x+1)OH^{-} = Ca_3Al_2(SiO_4)_{3-x}(OH)_{4x} + (11-5x)H_2O \tag{10}$$

It is possible that $H_7SiO_6{}^{-}$ is not the only component that yields hydrogarnite in the system. Another possible reaction is its formation via the chemical reaction (9), where the Si species is $HSiO_3{}^{-}$ [35]. However, $HSiO_3{}^{-}$ is not a stable component in the Eh-pH diagram compared to $H_7SiO_6{}^{-}$, but it is present in the HSC Chemistry database and was also considered by Nikolaychuk [34] in studying the Si-water system thermochemistry.

$$3Ca^{2+} + 2AlO_2{}^{-} + (3-x)HSiO_3{}^{-} + (x+1)OH^{-} + (2x-1)H_2O = Ca_3Al_2(SiO_4)_{3-x}(OH)_{4x} \tag{11}$$

A third explanation proposed is that the causticisation of the sodium aluminate solution by the addition of CaO results in the formation of calcium monocarboaluminate [36].

This, in turn, reacts with the Si in solution, forming hydrogarnite, thus acting as the actual desilicating agent [37].

It should also be noted that although tests no. 1 and 2 had no hydrogarnite as a product, they both still had some Si removal. The presence of Si in the solid residue from tests no. 1 and 2 was confirmed with XRF, but the phase could not be identified by XRD. This can be explained by the fact that hydroxides such as calcium monocarboaluminate are layered double hydroxides (LDHs) and can adsorb Si ions from solution and store them in their layered structure [38].

Lastly, the presence of $CaCO_3$ is likely due to the reaction between CaO and unreacted $NaCO_3$.

### 3.3. Precipitation of Aluminum Tri-Hydroxide

Only tests no. 1 and 2 produced the desired product, with the preferred Al tri-hydroxides being bayerite, gibbsite, and nordstrandite, and no traces of dawsonite (Table 10). Figure 14 is a sample XRD spectrum from test no. 1. The other three tests produced a product with a significant presence of dawsonite. In the cases of tests no. 3 and 4, this can perhaps be attributed to the precipitation and aging being conducted at elevated temperatures. A study [23] has shown that temperatures over 40 °C do favor dawsonite formation. In the case of test no. 5, this perhaps can be attributed to the use of a solution with a $Na_2CO_3$ concentration of 80 g/L. This has also been reported to favor dawsonite formation [24]. Regardless, going forward, it is recommended that all tests (carbonation and aging) be conducted with 60 g/L $Na_2CO_3$ concentration at temperatures less than 30 °C.

**Table 10.** Quantitative XRD analysis of phases in the precipitation filter cake (wt%).

| Solution Leach Test No. | 1 | 2 | 3 | 4 | 6 |
|---|---|---|---|---|---|
| Mineral Phase | | | | | |
| Bayerite [$\alpha$-Al(OH)$_3$] | 85 | 96 | 33 | 45 | 37 |
| Gibbsite [$\gamma$-Al(OH)$_3$] | 12 | 0 | 0 | 0 | 0 |
| Nordstrandite [Al(OH)$_3$] | 3 | 3 | 2 | 3 | 0 |
| Dawsonite [NaAlCO$_3$(OH)$_2$] | 0 | 0 | 65 | 52 | 67 |
| Natrite (Na$_2$CO$_3$) | 0 | 2 | 0 | 0 | 0 |

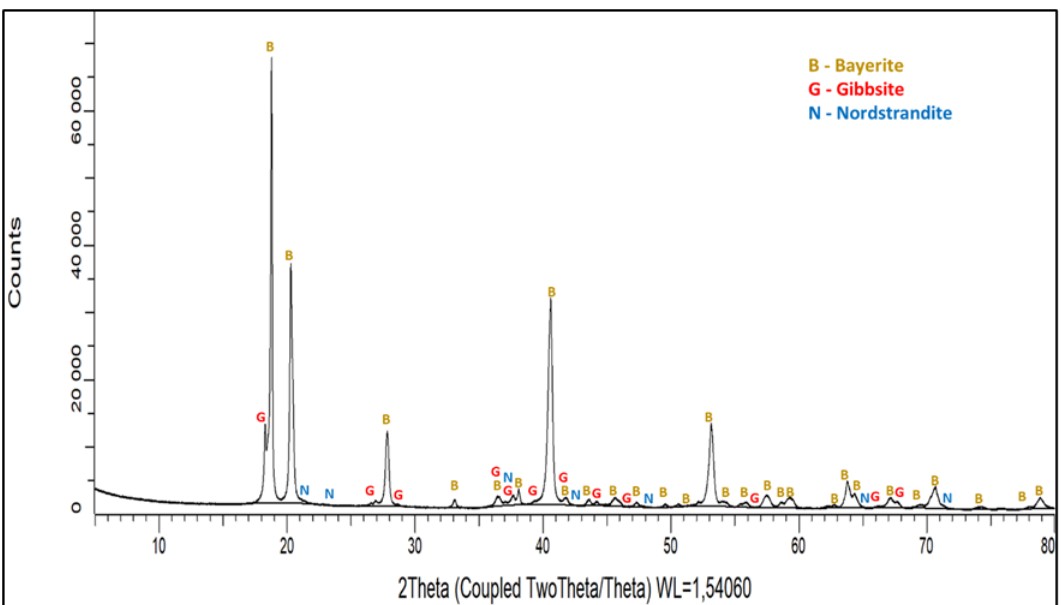

**Figure 14.** XRD spectrum for precipitate from test no. 1.

The precipitation of $Al(OH)_3$ occurs through the chemical reaction (3). The precipitation of dawsonite occurred at higher concentrations of $Na_2CO_3$, and this may be due to the presence of excess $Na^+$ ions according to reaction (10).

$$Al(OH)_4^-{}_{(aq)} + Na^+{}_{(aq)} + CO_{2(g)} = NaAlCO_3(OH)_{2(s)} + H_2O_{(l)} \tag{12}$$

XRF analysis of the precipitates (Table 11) shows that the presence of critical contaminants, specifically Si, Na, and Ca, was higher than typical industrial specifications [6]. Although the amount of $SiO_2$ in experiment 3 was 0.09% and an improvement from the other tests, where it was 0.55%, it was still higher than the required limit of 0.015%. The particle size distribution (PSD) analysis was only done for precipitates from experiments 1 and 2 as the others did not produce the desired products. Both products were far too fine to meet the specification of 92% > 44 µm [6]. Test no. 1 produced a product that was 90% < 35 µm, and test no. 2 a product that was 75% < 41 µm.

**Table 11.** XRF analysis of major contaminants in precipitation filter cake (wt%).

| | **Industrial** | **Test No.** | | | | |
|---|---|---|---|---|---|---|
| **Contaminant** | **Specification** | **1** | **2** | **3** | **4** | **6** |
| CaO | <0.04 | 0.09 | 0.20 | 0.27 | 0.13 | 0.20 |
| $SiO_2$ | <0.015 | 0.55 | 0.55 | 0.09 | 0.59 | 0.59 |
| $TiO_2$ | <0.004 | 0 | 0 | 0 | 0 | 0 |
| $Fe_2O_3$ | <0.015 | 0.01 | 0.03 | 0 | 0 | 0.02 |
| $Na_2O$ | <0.4 | 0.57 | 3.01 | 19.30 | 21.60 | 13.30 |

ICP analysis (Table 12) on the concentration of Al at the start and end of carbonation was conducted to determine the required time for precipitating all the Al in solution. Excluding test no. 3, because of the low concentration of Al at the start, a comparison was done by grouping tests no. 1 and 2 and comparing them with tests no. 4 and 6. The starting concentration of Al in tests 4 and 6 is nearly double that in 1 and 2. The carbonation time in 4 and 6 was 4 times that of 1 and 2, but the percentage of Al precipitated is similar for both groups. Regardless of starting concentration, the determining factor is the $Na_2CO_3$ concentration of the leaching solution. The higher the concentration, the longer the carbonation time required to precipitate all the Al.

**Table 12.** Amounts of Al in solution at the start and end of precipitation tests.

| **Test No.** | **1** | **2** | **3** | **4** | **6** |
|---|---|---|---|---|---|
| Start (g/L) | 12.04 | 9.88 | 5.38 | 20.9 | 18.30 |
| End (g/L) | 3.83 | 4.57 | 0.03 | 8.11 | 6.08 |
| Amount precipitated (%) | 68.19 | 53.74 | 99.44 | 61.20 | 66.78 |
| Carbonation time (h) | 0.5 | 0.5 | 1 | 2 | 2 |
| $Na_2CO_3$ leaching solution concentration (g/L) | 60 | 60 | 60 | 100 | 80 |

The process has shown promise and has a few areas that need to be further explored. The chemistry of the desilication process is yet to be understood and defined. Future work can also evaluate the use of $CO_2$ with higher partial pressure or possibly pure $CO_2$ to reduce the carbonation time. Alternatively, the use of a gas mixture like that of the furnace off-gas can be investigated in the precipitation stage. Due to the lower $CO_2$ component, the process will take longer, but it will be cheaper than using pure $CO_2$ gas that is purchased. Finally, seeding the solution with Al tri-hydroxide product in subsequent runs of the process may result in a product with the desired larger particle size.

## 4. Conclusions

This initial test program to evaluate the Pedersen process as an alternative route to producing alumina from bauxite residue has shown there is potential in the process, although several aspects must be improved and further understood. The study has shown that:

- Al occurred in the form of mayenite and gehlenite in the slag produced from bauxite residue. Moreover, in the leaching step, mayenite was almost completely leached.
- Although leaching resulted in Al extractions, no higher than 68% regardless of the slag mineralogy, it must be noted that this may not be a prohibiting factor when processing secondary and waste materials.
- The benefit of an intermediate mechanical step before re-leaching the material was minimal, while a solution concentration of 60 g/L of $Na_2CO_3$ appears to be ideal. Higher concentrations appear to cause passivation to occur faster.
- Desilication with CaO resulted in up to 88% of silica being removed with higher CaO additions. A mechanism for the desilication by CaO was proposed in which hydrogarnite is formed and deposited.
- The precipitation process was the most successful among the three processes, producing a product with the desired Al tri-hydroxides. The tests have shown that this is achieved by a combination of carbonation and aging done at temperatures lower than 30 °C on a leach solution with no more than 60 g/L $Na_2CO_3$ concentration.

**Author Contributions:** Conceptualization, J.M.M.; Methodology, J.M.M.; Software, J.M.M.; Validation, P.D.; Formal analysis, J.M.M., J.S.; Investigation, J.M.M.; Resources, J.S.; Data curation, J.M.M.; Writing—original draft, J.M.M. and J.S.; Writing—review & editing, J.M.M., M.V. and D.M., Visualization, J.M.M.; Supervision, J.S.; Project administration, J.M.M.; Funding Acquisition, J.S. All authors have read and agreed to the published version of the manuscript.

**Funding:** This project has received funding from the European Union's Horizon 2020 research and innovation program under grant agreement 767533. We also thank the partners in the ENSUREAL consortium for their continued support.

**Institutional Review Board Statement:** Not applicable.

**Informed Consent Statement:** Not applicable.

**Data Availability Statement:** Not applicable.

**Conflicts of Interest:** The authors declare no conflict of interest.

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
