# Peer review of "Investigating Aluminum Tri-Hydroxide Production from Sodium Aluminate Solutions in the Pedersen Process"

_processes, doi:10.3390/pr10071370_

Round 1

Reviewer 1 Report

The authors investigate the synthesis of alumina from low-grade feed materials over the Pedersen process. High Al recovery is obtained and aluminum tri hydroxide is formed at low temperatures at a fast rate. the results is very interesting and can be published on Processing at present.

Author Response

Thank you for your comments.

Reviewer 2 Report

This paper is investigating the industry-interested important alumina recovery process. The paper’s logic is clear. However, the discussion into some results and details are not satisfactory enough. Given all my comments below, I suggest a major revision before acceptance.

1. Fig. 7 and 9, for test no. 2 and no.6, why there is non-linearity in the leached Al curve?
2. In the introduction, the usage of high-grade alumina is not clear. Why we need it? E.g., for nanocomposites, etc.? The following papers could be of help:

[1] Understanding and designing metal matrix nanocomposites with high electrical conductivity: A review. S Pan, T Wang, K Jin, X Cai. Journal of Materials Science, 1-37

[2] Azof, F.I., Vafeias, M., Panias, D. and Safarian, J., 2020. The leachability of a ternary CaO-Al2O3-SiO2 slag produced from smelting-reduction of low-grade bauxite for alumina recovery. Hydrometallurgy, 191, p.105184.

The authors should discuss more about the applications and impacts of this process and the products.

3. Some details regarding the experimental process is missing:

What is the X-ray source ofr XRF, XRD, etc.? Cu alpha or?

What is the scanning rate for XRD?

What is the acceleration voltage for SEM and EMPA?

What is the solution you used for ICP-MS?

4. Fig. 2, Fig. 13, and Fig. 14: Better add the corresponding peak index. Also, what data base and reference card you used to match the products?

5. Fig. 6: Linearity should be assessed with R2-value…If lower than 80%, I do not think linear fitting is suitable.

Author Response

  1. Fig. 7 and 9, for test no. 2 and no.6, why there is non-linearity in the leached Al curve?

The following text was added in page 11 from line 299:

“It is not known definitively why the curve for test no.2 is non-linear as the samples were tested with two different labs and the pattern remained consistent. The outlier points could be interference caused by the additional amount of Na in the ICP-MS analysis. Further, it may have been precipitation of Al at those specific times in the reaction or it might have occurred in the sample bottles after the samples were taken.”

  1. In the introduction, the usage of high-grade alumina is not clear. Why we need it? E.g., for nanocomposites, etc.? The following papers could be of help:

[1] Understanding and designing metal matrix nanocomposites with high electrical conductivity: A review. S Pan, T Wang, K Jin, X Cai. Journal of Materials Science, 1-37

[2] Azof, F.I., Vafeias, M., Panias, D. and Safarian, J., 2020. The leachability of a ternary CaO-Al2O3-SiO2 slag produced from smelting-reduction of low-grade bauxite for alumina recovery. Hydrometallurgy, 191, p.105184.

The following text was added in page 1 line 2 in the Introduction:

Alumina is mainly used in the production of aluminium metal through high temperature electrolysis and must be of high purity. The other application of alumina is making metal matrix composites.

The authors should discuss more about the applications and impacts of this process and the products.

The following text in page 2

From lines 46-52:

“However, improvement in the smelter technology, the need for more sustainable processes in the mining industry and the need to address the red mud waste product from the Bayer process have renewed interest in this process. The process addresses all these matters while potentially being a zero-waste process producing by-products that can be used in other industries. Further, the Pedersen process has the potential to be used to remediate the red mud problem resulting from processing high grade bauxite ores through the Bayer process.  “

In line 58: The pig iron can potentially be processed to make cast iron.

In lines 62-66: In the second stage, the slags are leached with sodium carbonate solution to dissolve the aluminate minerals (reactions 1 and 2) and leave behind a mostly calcium carbonate product referred to as grey mud. Unlike red mud this by-product is environmentally benign and may be used as the principal feed material in the production of cement or fertilizer.

  1. Some details regarding the experimental process is missing:

What is the X-ray source ofr XRF, XRD, etc.? Cu alpha or?

The following has been added in page 9 line 237-238:

XRD analysis was conducted using Bruker D8 A25 DaVinci™ X-ray Diffractometer with CuKα radiation, 10 to 75 deg. diffraction angle, and scanning speed 0.01 deg.

What is the scanning rate for XRD?

The scan step was 0.01° which should correspond to a speed of 1 deg/min

What is the acceleration voltage for SEM and EMPA?

The following text was added on page 9 line 241-243:

EMPA was conducted on samples in polished resin blocks using a JEOL JXA- 8500F Electron Probe Micro Analyzer (EPMA) with Wavelength Dispersive X-ray Spectrometer with acceleration voltage 15.0 kV.

What is the solution you used for ICP-MS?

The following text was added on page 9 line 239-240:

“Samples for ICP-MS from leaching, desilication and precipitation tests were diluted in 15 ml Millipore water with 7 drops of 0.1 M nitric acid in preparation for analysis.”

  1. Fig. 2, Fig. 13, and Fig. 14: Better add the corresponding peak index. Also, what data base and reference card you used to match the products?

It is not possible to copy the corresponding peak index in a photo or any form. We can only copy the peaks and label as in the Figures.

The following text was added in page 9 line 238-240:

EVA and TOPAS software were used to interpret data against the Crystallography Open Database.

  1. Fig. 6: Linearity should be assessed with R2-value…If lower than 80%, I do not think linear fitting is suitable.

The figure is not meant to show the regression but rather the trend in Al recovery in relation to solution concentration.

The following text was added in page 10-11 line 291-293:

It is worth mentioning that the trendlines are not showing linear regression, however they are added to indicate the average changes of Al recovery. Moreover, this figure shows that the leaching recovery change with time is larger when lower solution concentrations are used.

Round 2

Reviewer 2 Report

With the better impact statement of this manuscript, it could be accepted.